# Application of Polypyrrole-Based Electrochemical Biosensor for the Early Diagnosis of Colorectal Cancer

**DOI:** 10.3390/nano13040674

**Published:** 2023-02-09

**Authors:** Xindan Zhang, Xiao Tan, Ping Wang, Jieling Qin

**Affiliations:** Tongji University Cancer Center, Shanghai Tenth People’s Hospital, School of Medicine, Tongji University, Shanghai 200092, China

**Keywords:** polypyrrole, electrochemical biosensor, biomarkers, early diagnosis, colorectal cancer

## Abstract

Although colorectal cancer (CRC) is easy to treat surgically and can be combined with postoperative chemotherapy, its five-year survival rate is still not optimistic. Therefore, developing sensitive, efficient, and compliant detection technology is essential to diagnose CRC at an early stage, providing more opportunities for effective treatment and intervention. Currently, the widely used clinical CRC detection methods include endoscopy, stool examination, imaging modalities, and tumor biomarker detection; among them, blood biomarkers, a noninvasive strategy for CRC screening, have shown significant potential for early diagnosis, prediction, prognosis, and staging of cancer. As shown by recent studies, electrochemical biosensors have attracted extensive attention for the detection of blood biomarkers because of their advantages of being cost-effective and having sound sensitivity, good versatility, high selectivity, and a fast response. Among these, nano-conductive polymer materials, especially the conductive polymer polypyrrole (PPy), have been broadly applied to improve sensing performance due to their excellent electrical properties and the flexibility of their surface properties, as well as their easy preparation and functionalization and good biocompatibility. This review mainly discusses the characteristics of PPy-based biosensors, their synthetic methods, and their application for the detection of CRC biomarkers. Finally, the opportunities and challenges related to the use of PPy-based sensors for diagnosing CRC are also discussed.

## 1. Introduction

As a result of the influences of individual lifestyle, food safety, and the ecological environment on health and the accelerating processes of industrialization, urbanization, and global population aging, the morbidity and mortality of chronic diseases are increasing with a constant upward trend, seriously endangering population health. Colorectal cancer (CRC) has one of the highest incidences among malignant tumors all over the world, which is related to genetics, gender, age, race, living environment, lifestyle, eating habits, and medicine use [1]. CRC develops from genetic and epigenetic variation and progresses to adenoma and malignancy through subsequent changes, such as transcription, translation, and abnormal protein expression, in a multi-stage process. There were over 1.9 million new CRC cases and over 900,000 deaths worldwide in 2020 [1]. Early diagnosis of CRC can enhance the survival rate with successful treatment and improve CRC outcomes by offering care at the earliest possible stage.

Recently, many techniques have been developed to diagnose CRC clinically, such as colonoscopy, sigmoidoscopy, colon capsule endoscopy (CCE), computed tomography colonography (CTC), the antibody-based fecal immunochemical test (FIT), the immune-based fecal occult blood test (FOBT), and biomarker determination [2]. Colonoscopy and sigmoidoscopy (endoscopic examination of the distal colon) are practical tools for CRC screening with higher sensitivity that can clearly show lesions, and they can be utilized as auxiliary options during CRC surgery. However, these methods are invasive and costly and require special facilities and sedatives with low patient compliance [3,4]. CCE is a potential noninvasive screening tool for CRC, but it is difficult to control and lacks accuracy in the determination of the type and size of lesions [5]. CTC is an auxiliary imaging method that can be used to identify colonic lesions in three-dimensional (3D) images with high sensitivity, and it is less invasive and suitable for visualization of the entire colon [6,7]. Nevertheless, CTC is also limited due to the uncomfortable bowel preparation, radiological safety concerns, and poor specificity in the detection of small tumors [8]. Standard non-invasive assays, such as FIT and FOBT, enable the quantitative measurement of hemoglobin content in human fecal samples [9,10]. However, they lack specificity in the detection of precancerous lesions and are related to false positives in clinical tests [11]. In contrast, the detection of CRC biomarkers, including Kirsten rat sarcoma viral oncogene homolog (KRAS), V-RAF murine sarcoma viral oncogene homolog B1 (BRAF), tumor protein 53 (TP53), microRNA-21 (miRNA-21), carcinoembryonic antigen (CEA), carbohydrate antigens (CA19-9, CA72-4, CA125), interleukin-6 (IL-6), and vascular endothelial growth factor (VEGF), is more sensitive and has important practical significance for the early screening of CRC [12,13].

Biomarkers mainly refer to biological molecules—i.e., DNA, RNA, miRNA, and proteins—that are markers of normal or abnormal states [14]. The conventional detection methods include the polymerase chain reaction (PCR) method [15], the enzyme-linked immunosorbent assay (ELISA) [16], DNA microarrays [17], Northern blot techniques [18], electrophoresis [19], radioimmunoassays (RIAs) [20], immunohistochemistry [21], chromatography-based technologies [22], the fluorescence method [23,24], and the chemiluminescence method [25]. Although these methods have been used to obtain relatively accurate results, detecting trace biomarkers at the early stage of CRC is far from effective. Moreover, the limitations include complex operation, a time-consuming detection process, the need for high-level technology and expensive equipment, and low sensitivity. Thus, there is still a strong demand for specific tools for the early detection of CRC that are efficient, sensitive, accurate, convenient, and fast.

Electrochemical-based detection methods, including electrochemistry, electrochemiluminescence (ECL), and photoelectrochemistry (PEC) methods, can convert analytical signals generated by target molecules into readable electrical signals, and they have the inherent advantages of excellent sensitivity and selectivity, simple operation, rapid detection, cost-effectiveness, simultaneous detection of multiple biomarkers, and potential for miniaturization [26]. Electrochemical techniques can be divided into five categories in accordance with the measurable signals: amperometry, potentiometry, resistance methods, voltammetry, and conductometry (Figure 1). Among them, voltammetry and electrochemical impedance spectroscopy (EIS) are often used for the construction of electrochemical biosensors, sensor characterization, and quantitative analysis [27,28]. Voltammetry has the advantages of simple operation, intuitive atlas analysis, widespread effectiveness, and high sensitivity. EIS has no significant influence on the system during the measurement process. It provides more interface structure and electrode dynamics information than other electrochemical methods.

Electrochemical sensors are sensitive sensing devices that use electrochemical technology to detect analytes; they are generally composed of sensitive elements, transducers, and transformation circuits. Electrochemical sensors can be categorized according to their different target molecules as biosensors, gas sensors, ion sensors, etc. The biometric element is a sensitive element with a molecular recognition ability that has a significant place in the construction of biosensors. In accordance with the different biometric elements, biosensors can be divided into enzyme sensors, immune sensors, aptamer sensors, and so on. Moreover, the electrode reaction in the sensor occurs at the interface between the solution and the electrode, and the properties of the interface have a significant influence on the reaction. Therefore, reasonable modifications of the interface play a decisive role in improving sensing efficiency.

Nanomaterials and nanocomposites with different structures [29,30], including conductive polymers (CPs), carbon-based materials, metal nanomaterials, metal oxide nanomaterials, and silicon nanomaterials, are often utilized in biosensors to increase the surface area, fix the biometric elements, catalyze the electrochemical reactions, enhance the electrical conductivity of the electrode surface, and label the biomolecules, improving the detection performance of the sensors.

Among the various nanomaterials, increasing attention has been focused on CPs, which are known as “synthetic metals” because of their outstanding electrical, optical, and magnetic characteristics [31]. Furthermore, several CPs, such as PPy, polyaniline (PANI), polythiophene (PTh), and poly(3,4-ethylene dioxythiophene) (PEDOT), have gained extensive attention for practical applications relating to biomedicine, electronics, energy equipment, etc. because of their biocompatibility, high surface area, good environmental stability, inherent electrical conductivity, and other physical properties [32,33]. Specifically, these CPs can be used for a wide range of applications in electrochemical biosensors, as shown in Table 1. Wang et al. utilized polypyrrole nanowires (PPyNWs) and polyamidoamine dendrimer (PAMAM) to design an miRNA biosensor with a high surface area and high electrical conductivity that showed significantly improved sensitivity in the determination of miRNA [34]. Compared to other CPs, PPy has been widely studied and applied, especially for the development of implantable, flexible, and wearable electronic equipment, due to its electrical versatility, which ranges from that of an insulator to near that of a metal; outstanding optical, thermoelectric, and electrical characteristics; easy synthesis and functionalization; low electropolymerization potential; stability under environmental conditions; and biocompatibility [35,36,37].

## 2. Polypyrrole Biosensors

### 2.1. Physical and Chemical Characteristics of PPy

Pyrrole monomer is a five-membered heterocyclic molecule composed of C and N that appears as a colorless oil-like liquid at room temperature. PPy, a heterocyclic conjugated conductive polymer that usually appears as an amorphous black solid with good electroconductivity, processability, and chemical stability, is easy to form through polymerization of pyrrole in various organic electrolytes. Although conventional PPy has high rigidity, poor mechanical ductility, and poor solubility in common organic solvents, as well as deficiencies in its optical, electrical, and biological properties, nanostructured PPy has improved electrochemical activity, better electrical conductivity and biocompatibility, enhanced optical properties, good mechanical properties, and is easy to process because of the nanostructure and larger surface area, making it widely utilized in biomedical applications [51,52].

### 2.2. Synthesis and Modification of PPy

The polymerization of pyrrole can be carried out using chemical, electrochemical, ultrasonic, electrospinning, and even biotechnological methods with different morphologies (Figure 2), among which chemical oxidation polymerization and electropolymerization are commonly used [53,54,55,56,57,58,59]. During the synthesis of PPy, the CP is able to carry biomolecules or functional groups for specific biometric functions through physicochemical means [60,61]; i.e., physical adsorption, embedding, affinity, covalent immobilization, etc.

Oxidative chemical polymerization of PPy is inexpensive and suitable for large-scale production. Andriukonis et al. proved that [Fe(CN)_6_]^3-^ can induce the synthesis of PPy [62]. Mao and Zhang pointed out that FeCl_3_, H_2_O_2_, and other oxidants can be utilized to oxidize Py and polymerize PPy [63]. Furthermore, pyrrole monomer can also be oxidized using oxidoreductases (e.g., peroxidase, glucose oxidase, etc.) through enzymatic reactions in an environmentally friendly fashion with a suitable pH and room temperature conditions [64,65], and the embedded enzyme can maintain its catalytic activity when encapsulated in the polymer particles or layers formed during the preparation of enzyme-based biosensors. In addition, the formation of PPy can be implemented in cells, where the PPy induced by microorganisms is mainly deposited in the cytoderm and between the cell membrane and the cytoderm [66,67].

Electrochemical polymerization can be used to form PPy in situ with or without embedding materials (nucleic acids, enzymes, receptor proteins, and antibodies) to improve the mechanical properties and solubility [34,68,69,70,71]. In the electrochemical preparation of PPy, selection of the appropriate electrodeposition methods and electrochemical parameters, including the applied voltage, current, potential window, potential scanning rate, and duration, can ensure that the morphology, thickness, conductivity, doping, and dedoping of the PPy are regulated well [72]. Moreover, the electrochemical properties of PPy, such as the conductivity, morphology, thickness, structure, and porosity, can also be modulated by controlling the type and concentration of dopants, electrolytes, and solvent, as well as the pH value, temperature, and monomer concentration of the bulk solution [54,73,74]. In addition to normal PPy preparation, electropolymerization can be utilized for the synthesis of molecularly imprinted polymer (MIP) films on the electrode surface with high stability and low cost [72,75,76]. During the preparation of the MIP, additional electrochemical operations can be performed for overoxidation after the formation of the initial electrodeposited PPy layer [77,78]. Although peroxidation will destroy the π–π conjugate system of CP and inhibit the polymerization process, during the construction of the MIP, oxygen-containing groups, such as hydroxyl (-OH), carbonyl (-CH = O), and carboxyl (-COOH), can be generated adjacent to the embedded molecules, forming a specific environment conducive to the attachment of imprinted template molecules. Moreover, excessive oxidation is able to promote template removal and regeneration based on the MIP layer.

In addition to the direct synthesis of PPy via chemical oxidation and electrochemical oxidation, ultrasound can also be used to promote the polymerization of Py. The ultrasonic cavitation effect produced in the ultrasound process can heat the solvent and atomize it locally, making it possible to prepare polypyrrole with a small size and uniform shape in a short time [79]. Electrospinning is another convenient method that can be used to produce ultra-fine polymer nanofibers with a porous structure and high specific surface area [80]. Vapor phase polymerization does not require a solution environment and can be used to produce high-purity PPy nanomaterials on different types of substrates. However, the formed PPy has insufficient adhesion with the substrate surface [81]. Photopolymerization is a technology with which Py monomers can be polymerized under visible light or ultraviolet light, using laser-generating free radicals for the preparation of porous PPy with a high specific surface area. This approach allows for excellent control of the size of PPy and facilitates micro-machining and direct polymerization on substrates with solubility or temperature limitations [82].

### 2.3. Applications of PPy

As shown in Figure 3, PPy—with different morphologies including nanoparticles, nanotubes, nanowires, nanorods, nanocapsules, thin films, nanofibers, and hydrogels—can be utilized as a conductive material, electrical display material, electrochromic material, or photoluminescent quencher [35,83,84] in the construction of chemical sensors, biosensors, optical sensors, actuators, flexible electronics, transistors, electrochemical batteries, biofuel cells, photovoltaic cells, electrochromic displays, wearable and implantable/connectable biomedical tools, and other sensors [31,85,86,87,88,89,90,91,92,93,94]. Yang et al. prepared a high-performance biosensor utilizing PPy nanowires (PPyNWs) with outstanding conductivity and a large surface area to detect hydrogen peroxide and miRNA [95]. Rong et al. reported a nanocomposite film containing PPy for the selective adsorption and removal of Pb [96]. Mohamed et al. designed a nanocomposite using PPy nanofibers (PPyNFs) with photocatalytic activity and the capacity for dye adsorption to remove dye from raw water samples [97]. Han et al. fabricated a conductive hydrogel combining PPy and silk hydrogel for use in the preparation of flexible and wearable sensors [91]. Fan and his colleagues utilized a PPy sponge with micron-sized pores, good mechanical capacities, and the capacity for light absorption for use in a functional solar steam generator [98].

### 2.4. Application of PPy-Based Biosensors

PPy and its derivatives are some of the most effective nanomaterials for improving the sensing performance of different biosensors (see Table 2). PPy nanomaterials and their composites, which have unique optical, electrochemical, and other physical and chemical properties, have great potential for the enhancement of sensing performance, including the response and recovery time, stability, selectivity, and sensitivity [99].

#### 2.4.1. Enzyme-Based Biosensors

PPy is widely used as a substrate for the preparation of enzyme-based biosensors. During this process, PPy is used for enzyme fixation and facilitates electron transfer between the active center of the immobilized enzyme and the electrode. The permeability of the PPy layer for enzyme substrates and reaction products is relatively low, contributing to the increase in the apparent Michaelis constant of the enzyme and thereby expanding the detection range for analytes [115]. Apetrei et al. described a PPy-based enzymatic sensor in which the polyphenol oxidase extract could be utilized as a catalyst for both the synthesis of PPy and the self-encapsulation of the enzyme (Figure 4A) [100]. Dutta et al. synthesized an amperometric biosensor in which PPy was used to entrap and immobilize acetylcholinesterase, facilitate electron transfer, and reduce the oxidation potential of the reaction substrate [101]. Shi et al. synthesized overoxidized PPy to modify an electrode, and it had a catalytic oxidation ability that made it possible to reduce the oxidation potential of ascorbic acid and then increase the sensitivity of the biosensor [102].

#### 2.4.2. Immunobiosensors

Antibody/antigen-based biosensors are sensing devices based on the affinity interaction between antibodies and antigens [116]. PPy is commonly applied in these sensors because of its low oxidation potential and good biocompatibility. During biosensor preparation, PPy can serve as both an immobilizing substrate for biometric components and as a signal transduction system [99]. Tang et al. fabricated PPy-PEDOT-Au to fix the antibody in an electrochemical immunosensor due to the excellent electron transfer efficiency and environmental stability of PPy (Figure 4B) [104]. Zou et al. utilized PPy with good biocompatibility and a high surface area to improve the dispersion of AuNPs, constructing a biocompatible platform and immobilizing more antibodies to detect trace amounts of *E. coli* K12 (Figure 4C) [105].

#### 2.4.3. Aptamer-Based Biosensors

Aptamers, which are artificial single-stranded oligonucleotide fragments, can precisely capture ligands, such as DNA, RNA, or proteins, with high affinity and selectivity. Aptamers are easily synthesized and more stable than antibodies and antigens, and they have been widely utilized in electrochemical biosensors to detect low concentrations of target molecules in the blood. Duan et al. designed an aptasensor to detect lipopolysaccharide in which PPyNWs with -COOH ensured the immobilization of aptamers and enlarged the specific surface area (Figure 5A) [106]. In addition, a nitrogen-doped graphene (NG)/PPy nanocomposite was synthesized in order to design an aptasensor that, with the help of PPy, would exhibit a large surface area and enhanced electroconductivity [107].

#### 2.4.4. MIP-Based Biosensors

MIP is synthesized by polymerizing one or more monomers in the presence of the template molecules and then removing the template analytes, forming a complementary 3D imprint in the polymer matrix. It is a kind of polymer with specific recognition sites that are created artificially and complement the imprinted analyte, and it is also known as a “bionic receptor” [117]. Considering its biocompatibility, excellent electron transmission rate, and good environmental stability [108,110], molecularly imprinted polypyrrole (MIPPy) can be polymerized and impressed simultaneously using amperometric, potentiometric, or potential scanning methods or polymerized on a template-modified electrode [75]. It can be used for biomedical and environmental monitoring applications, and it is cost-effective and exhibits excellent selectivity, sensitivity, and chemical/thermal stability [75,111]. Ding et al. described an electrochemical method that can be used to detect glyphosate with a detection limit of 1.94 ng/mL based on the construction of molecularly imprinted polypyrrole nanotubes (MIPNs) that exhibited excellent conductivity and specificity (Figure 5B) [108]. Wu et al. designed an electrochemical platform utilizing imprinted polypyrrole from bacteria that could sensitively detect *E. coli* O157. Owning to the presence of MIPPy, the biosensor had high selectivity and specificity and a rapid detection ability and was easy to prepare (Figure 5C) [109]. 

#### 2.4.5. Nanocatalytic Biosensors

As natural enzymes are limited by their instability, demanding environmental conditions, and complex preparation processes, nanocatalytic biosensors have been developed that provide improved electrocatalytic behavior, stability, and selectivity because of their satisfactory surface area, morphology, and high conductivity [118,119,120]. After Fe_3_O_4_ nanoparticles were reported to exhibit catalytic activities similar to enzymes in 2007 [121], a growing number of catalytically active nanomaterials have been discovered that can be used as alternatives to enzymes. Metals and metal oxides have outstanding electronic properties, and their nanoenzymes have received significant attention for sensing and electrocatalysis applications [118]. Li et al. prepared nanocomposites capable of catalyzing glucose oxidation by depositing gold nanoparticles on the surface of PPyNFs to non-enzymatically detect glucose, using PPyNFs as carriers to avoid the need for harsh oxidation pretreatment and disperse the AuNPs well, thereby improving their catalytic activity and reducing Au consumption (Figure 6A) [112]. Jeong and his colleagues proposed an electrochemical method based on a chitosan–PPy/TiO_2_ sensor in which TiO_2_ NPs, as the electrochemical catalyst, are deposited using the plasma process, demonstrating outstanding catalytic activity, reactivity, sensitivity, and selectivity in the detection of glucose (Figure 6B) [113]. Meng et al. fabricated a non-enzymatic biosensor utilizing Cu_x_O-modified PPyNWs to sensitively detect glucose, and the PPyNWs showed excellent electrical performance and produced a good immobilization matrix for nanoparticles (Figure 6C) [114].

## 3. PPy-Based Biosensors for CRC Biomarker Detection

CRC is currently the second-leading cause of death worldwide [122]. The electrochemical detection of biomarkers in the blood (such as DNA, RNA, miRNA, proteins, and other molecules), a low-cost, simple, rapid, specific, sensitive, and noninvasive strategy, exhibits tremendous potential for the early diagnosis of CRC [27].

### 3.1. Circulating Cell-Free DNA (ccf-DNA)

Ccf-DNA is actively released by tumor cells through apoptosis, necrosis, or exosomes. It is present in significantly higher amounts than in healthy people, highlighting its crucial practical significance for the early diagnosis of malignant tumors. Ccf-DNA is attractive and easily accessible, providing a new, non-invasive method for CRC detection and characterization. Many studies have shown the diagnostic, predictive, and prognostic significance of abnormal ccf-DNA with genetic and epigenetic variations in the plasma/serum of CRC patients [123].

#### 3.1.1. DNA Mutation

Abnormal genetic mutations in the blood have been evaluated as one of the most promising diagnostic tools for CRC. The protooncogene KRAS is an early candidate in this context, as its mutations involve some of the most frequently mutated oncogenes and have a certain practical significance for the clinical diagnosis of CRC [122]. BRAF, belonging to the RAF gene family, is a direct downstream effector of KRAS and is connected to CRC development [123]. The adenomatous polyposis coli (APC) gene, a tumor-suppressor gene with the encoded protein in the Wnt signaling pathway, was one of the early genetic factors used in the screening of CRC [124]. It is claimed that TP53 is the guardian of the genome, and its mutation can disable the functional activity of wild-type p53 (wtp53) and produce oncogenic properties [125]. As the TP53 mutation occurs in 50–70% of CRC patients, monitoring the changes in the TP53 gene and/or its encoded protein in CRC patients may contribute to the early diagnosis and detection of clinical conditions.

As shown in Figure 7A, Wang and colleagues coated PPy-covered MWNT-Ru(bpy)_3_^2+^ composite materials on the surface of a Au electrode to prepare a DNA sensor for wtp53 sequence detection using ECL [126]. ECL, which has the advantages of low background signals, high sensitivity, good versatility and controllability, and a wide detection range, is generated using electrochemical reactions that trigger light signals. Considering its large specific surface area and prominent conductivity, PPy was utilized as a stable modification layer for ssDNA attachment to improve the wtp53 sensing performance, resulting in a detection range of 0.2 pM–200 pM and an LOD of 0.1 pM. The authors also fabricated another DNA-based biosensor modified by electrospinning composite MWNT-PA6-PPy nanofibers (Figure 7B) [127]. Compared with traditional planar materials, PPy nanofibers have better mechanical strength, uniformity, porosity, and reusability, as well as satisfactory biocompatibility and a high surface area, making it possible to immobilize more ssDNA and increase the hybridization sensitivity for determination of trace amounts of wtp53.

#### 3.1.2. DNA Methylation

DNA methylation is another ccf-DNA-based technique for the early diagnosis of CRC that employs genes such as SEPT9, SCTR, SDC2, SFRP2, TMEFF2, NGFR, and CG10673833 [128,129,130,131]. Sun et al. demonstrated that aberrant methylation of the SEPT9 gene (^m^SEPT9) in the blood can be used as a marker in the early diagnosis and screening of CRC [128]. Li et al. showed that hypermethylation of the SCTR gene resulted in good accuracy in the diagnosis of CRC and its precursor lesions [129].

### 3.2. MiRNA-Based Biomarkers

MicroRNAs are abundant and endogenous noncoding RNAs with a small size and hairpin structure. MiRNAs can regulate and control physiological processes, such as the proliferation and differentiation of cells, by regulating the expression of various genes. The development of numerous diseases (cancer, neurodegenerative diseases, cardiovascular diseases, etc.) is related to abnormal microRNA expression [132]. It has been shown that miRNAs play significant roles in the progress of CRC, and evaluation of the strange expression of miRNAs, such as miR-21, miR-92a, miR-451a, miR-29a, miR-23a, miR-141, let-7a, miR-1229, miR-223, miR-1246, miR-150, and miR-378, has shown great clinical value in CRC screening, prognosis, prediction, and treatment [133,134,135,136]. Among these miRNAs, miR-21 is one of the most widely researched for the diagnosis, prediction, and treatment of CRC [134,136,137,138]. Pothipor et al. synthesized a gold nanoparticle/polypyrrole/graphene (AuNP/PPy/GP) nanocomposite for the selective detection of miR-21 (Figure 8A) [139]. The use of PPy in this work improved the dispersion of AuNPs on the electrode surface, facilitating the fixation of an miR-21 probe with a corresponding detection range of 1.0 fM–1.0 nM and LOD of 0.020 fM that can be used to detect miR-21 in clinical trials. Tian et al. designed a PPy-AuNP superlattice (AuNS) biosensor for the detection of miR-21 (Figure 8B) [140]. Compared to randomly arranged nanoparticles, the presence of the conductive polymer PPy can induce AuNPs to assemble into AuNS structures with a larger surface area, better electron transfer performance, and more active sites. Furthermore, using PPy ligand can facilitate quantitative and accurate control of the distance between adjacent particles, enabling miR-21 determination with a 100 aM–1 nM detection range and limit of detection (LOD) of 78 aM.

Kaplan et al. sensitively detected miRNA-21 using electropolymerized PPy on the surface of a pencil graphite electrode (PGE) and achieved maximum doping of the anti-miR-21 probe in PPy (Figure 9A) [141]. PPy has high conductivity and a porous structure, making it possible to improve charge transfer, increase the number of fixed probe molecules, and reduce the non-specific binding of MDB and other molecules, and the designed biosensor demonstrated improved selectivity and an LOD of 0.17 nM. Yang et al. prepared PDA-PPy-NS with π-electron coupling and an ultra-narrow band-gap by polymerizing PPy onto hybrid polydopamine nanosheets (PDA-NSs) and used nucleic acid dye (Cy5)-labeled ssDNA as probes to detect miRNA-21 (Figure 9B) [83]. The mixed PDA-PPy-NS nanoquencher showed a better fluorescence quenching ability than PDA-NSs owing to the presence of the narrow band-gap PPy and its excellent π-electron delocalization ability, promoting intermolecular electron coupling and realizing fluorescence quenching. The nanoquencher/probe was demonstrated to have remarkable specificity, stability, and sensitivity and an LOD of 23.1 pM, indicating the tremendous potential of nanoquencher-based sensors for detection with real samples.

### 3.3. Specific Protein Biomarkers

In addition to DNA and RNA, various proteins secreted from tumor cells can also facilitate the early diagnosis of CRC, including CEA, CA19-9, CA72-4, CA125, IL-6, IL-8, MUC1, and p53, as shown in Table 3. As the table demonstrates, conductive PPy is commonly applied to improve electrochemical sensing performance, including the detection range and LOD, in the detection of specific CRC protein biomarkers.

#### 3.3.1. Carcinoembryonic Antigen (CEA)

The polymeric glycoprotein CEA is a tumor-associated antigen that is overexpressed in CRC, gastric cancer, breast cancer, lung cancer, and other cancers and has a certain value for the evaluation of tumor status and therapeutic effect [157]. CEA is abnormally expressed in more than 90% of CRC patients. The monitoring of CEA concentration in serum/plasma is an effective strategy for CRC diagnosis and measurement of disease progression. 

Tavares et al. designed a self-powered and self-signaled biosensing platform for the detection of CEA using MIPPy and the DSSC method (Figure 10A) [158]. Specifically, they assembled the MIPPy as a biorecognition element on the PEDOT layer of an FTO-conductive glass substrate and used it as the counter electrode of the DSSC. The DSSC/biosensor device was connected to an electrochromic cell to produce a color gradient for the CEA. The concentration of CEA ranged from 0.1 ng/mL to 100 μg/mL. This strategy can be used for clinical point-of-care (POC) analysis with high independence.

PPy can also be used to prepare flexible pressure sensors that convert external force information into electrical signals in real time. Yu et al. prepared a pressure-based immune sensor based on 3D PPy foam (Figure 10B) [143]. The PtNPs attached to dAb catalyze the decomposition of H2O2 and produce oxygen in the sealed device. Consequently, CEA concentration can be detected using pressure changes in the range of 0.2–60 ng/mL with an LOD of 0.13 ng/mL.

Zhu et al. constructed a novel ECL immune-based biosensor using anti-CEA–luminol–AuNP@PPy (Figure 11A) [144]. The PPy nanostructure made it possible to enhance the conductivity of the biosensor and provided a high specific surface area for the combination with the AuNPs, thus enabling the attachment of abundant amounts of the ECL reagent (luminol) and promoting the immobilization of antibodies. This strategy showed a detection range of 0.01 pg/mL–10 ng/mL and an LOD of 3 fg/mL, making it an effective tool for the clinical detection of CEA.

CP hydrogels, such as PPy hydrogel, have been applied to construct biosensors; in particular, those with 3D nanostructures. Compared to other materials, PPy hydrogel has the advantages of prominent conductivity, good biocompatibility, a large specific surface area, and easy processing [147]. Rong et al. designed an electrochemical immunosensing platform to measure CEA based on 3D continuous conducting network nanocomposites composed of PPy hydrogel loaded with AuNPs (Figure 11B) [147]. The hydrogel had good biocompatibility and electronic properties and offered a larger space that made it possible to immobilize more biomolecules. The nanostructure-based sensor had a highly porous 3D network, high specificity, and good stability. It showed a broad linear range (1 fg/mL–200 ng/mL) and an LOD of 0.16 fg/mL.

Abnormal changes in CEA concentration in the blood are generally associated with cancer progression, and the sensitivity of this biomarker increases with tumor stage. Therefore, CEA is the preferred marker for monitoring CRC progression and prognosis. However, abnormal elevation of CEA in the blood occurs not only in CRC but also in various other diseases. In addition, CEA has low sensitivity in the early stages of CRC. Hence, this biomarker is ineffective for screening and detecting CRC early [123].

#### 3.3.2. Carbohydrate Antigens

Abnormal expression of CA19-9, CA72-4, CA125, CA242, and other carbohydrate antigens also has a certain correlation with CRC [157,159,160,161]. CA19-9 was first discovered in 1981, and elevated CA19-9 levels can be used both as an aid in the diagnosis of CRC and as a reference index to assess the development of CRC. 

CA72-4 is a cancer biomarker with a certain diagnostic value that can provide diagnostic information regarding recurrent CRC. Lv et al. proposed an ECL immunosensor using a novel Ag2Se@CdSe nanomaterial nanoneedle modified with polypyrrole-intercalated aminated graphene (PPy-NH_2_GO) for CA72-4 detection (Figure 12A) [148]. The PPy-functionalized NH_2_GO had a high surface area that made it possible to immobilize significant amounts of Ag_2_Se@CdSe, and it demonstrated a low LOD of 2.1 × 10^−5^ U/mL and a detection range of 10^−4^–20 U/mL.

An electrochemical magnetic immunoassay platform was described by Huang et al. that used anti-CA125 antibody (mAb_1_)-conjugated magnetic beads as the capture probe and an anti-CA125 antibody (pAb_2_)-labeled Ag-PPy nanostructure as the detection probe (Figure 12B) [149]. Compared with AgNPs, use of Ag-PPy as the electroactivity indicator can further improve sensors’ analytical performance.

However, due to the limited applicability of markers such as CEA and CA19-9, several other proteins have been highlighted as potential biomarkers associated with CRC.

#### 3.3.3. Interleukin-6 (IL-6)

IL-6 is an inflammatory cytokine with hematopoietic and immunomodulatory functions; although not a specific biomarker of CRC, IL-6 is closely associated with CRC occurrence, development, staging, invasion, and metastasis.

Tertis et al. constructed an electrochemical aptasensor to sensitively detect IL-6 in human serum by depositing a nanocomposite consisting of PPy nanoparticles (PPyNPs) and AuNPs onto SPCE (Figure 13A) [151]. Both the PPyNPs and AuNPs coexisted on the electrode surface, providing an appropriate environment for immobilization of IL-6 aptamers. Moreover, conjugated polypyrrole polymer-containing epoxy side group (PPCE) is a novel conjugated polymer polymerized by the Py monomer that contains an epoxy active side group. As shown in Figure 13B, Elif Burcu Aydın designed an immunosensor using PPCE-modified ITO electrodes [152]. The PPCE polymer formed a large specific surface area on the ITO electrode, making it possible to fix the biomolecular IL-6 receptor, and the sensor had good conductivity, stability, and biocompatibility, which enhanced its sensitivity.

PPyNWs exhibit good electrical characteristics and can enhance the sensitivity of biosensors because of their high specific surface area. Cruz et al. prepared PPyNWs on PEEK and PETE flexible thermoplastics using nanocontact printing technology and controlled chemical technology and then functionalized them with diazo chemistry and a crosslinking agent to immobilize the IL-6 antibody, enabling IL-6 detection with a wide linear range from 1 pg/mL to 50 pg/mL and an LOD of 0.36 pg/mL [154].

#### 3.3.4. Vascular Endothelial Growth Factor (VEGF)

Angiogenesis is closely connected to the growth of solid tumors and the metastasis of cancer cells, and VEGF participates in the regulation of angiogenesis by stimulating the corresponding receptors and is important for the development of blood vessels. Nogués et al. pointed out that high-level expression of VEGF in the serum of CRC patients seems to be a promising tumor biomarker [162]. 

By immobilizing anti-VEGF RNA aptamers onto a field-effect transistor (FET) modified with carboxylated polypyrrole nanotubes (CPNTs) with excellent conductivity, Kwon et al. developed a biosensor capable of recognizing VEGF, as shown in Figure 14A [155]. The FET platform is capable of detecting VEGF at concentrations as low as 400 fM. In a related study, the author led a team in combining PPy-transformed N-doped few-layer graphene (PPy-NDFLG) with an RNA aptamer for the fabrication of a high-performance and flexible FET biosensor (Figure 14B) [156]. The sensor demonstrated good mechanical flexibility, high sensitivity, high selectivity, a rapid response, reusability, durability, and a low LOD of 100 fM.

#### 3.3.5. Other CRC-Related Protein Biomarkers

Mucin 1 (MUC1) is a membrane-associated macromolecule glycoprotein that is overexpressed in most adenocarcinomas [163,164]. Detection of abnormal increases in MUC1 in the blood can offer new opportunities for CRC early diagnosis, tumor staging, and clinical treatment.

Huang et al. designed a microfluidic aptasensor by combining the use of an MUC1 aptamer as a detection probe with PPyNWs. The sensor can be used for sensitive, rapid, label-free, and real-time detection of MUC1 [165]. The PPyNW-modified biosensor showed significantly enhanced sensitivity, conductivity, and biocompatibility. In addition, serum angiopoietin, MST1/STK4, S100A9 TIMP1, ITGB4, Cyr61, and CXCL-8 in the blood have also been proven to have potential uses as biomarkers in the diagnosis and detection of colorectal cancer, and they could be employed in electrochemical sensor detection after further verification [166,167,168,169,170].

### 3.4. Opportunities and Challenges Related to PPy-Based Sensors in CRC Diagnosis

Currently, most discovered CRC biomarkers are not specific to CRC. Combined detection of multiple biomarkers in the blood could sufficiently increase the accuracy of CRC diagnosis, resulting in an effective detection strategy. It has been shown that, compared with the detection of single biomarkers, simultaneous determination of the serum biomarkers MMP-7, TIMP-1, and CEA increased the sensitivity and specificity of CRC diagnosis [171]. However, studies are still required to further determine the clinical significance of single CRC marker analysis and combined detection of groups of biomarkers as early detection tools for CRC and to develop additional analytical means that can enhance the accuracy and specificity of CRC diagnosis. 

As a heterocyclic conjugated polymer with good electroconductivity, processability, and chemical stability, PPy is rarely used for the detection of multiple CRC biomarkers. The electrochemical deposition of PPy could enable sensing of coating designs with different physical characteristics and the development of arrays of electrochemical biosensors in which a single sensor would respond differently to similar mixtures of the analyte, opening applications for the detection of CRC biomarkers. Moreover, the biocompatibility of PPy provides it with potential for use in the design of implantable biomedical devices. However, PPy lacks selectivity for target molecules, and the modification of suitable biometric molecules is essential. In addition, while PPy may not always show the best results alone in an electrochemical biosensor, copolymers blended with other CPs could also be applied for suitable trace sensing. Moreover, PPy may degrade over time during detection, and efforts should be devoted to improving the stability of sensor response. In addition, in past decades, researchers mainly focused on the interfacial design, employing different PPy polymerization methods to improve the stability of the enzymes. However, research on the interface interaction between PPy and biorecognition elements, which has the potential to improve the sensitivity of biosensors, is lacking. Therefore, the interface interaction between PPy and biorecognition elements should be focused on and discussed in future studies with the aim of increasing sensing performance.

## 4. Conclusions

CRC is a type of cancer with high morbidity and mortality worldwide. Analyzing the concentration changes in CRC-associated biomolecules through electrochemical biosensing technology is significant for the early diagnosis, prognosis, and prediction of CRC. The fundamental purpose of this review paper was to introduce the design of PPy biosensors and their applications for the electrochemical measurement of CRC biomarkers. The conductive polymer PPy is a nanomaterial that has attracted much attention because of its specific characteristics, such as its excellent electrical properties, the flexibility of its surface properties, its easy preparation and functionalization, and its good biocompatibility, and it is often used in the design and improvement of biosensors. The different forms of PPy, synthesized via oxidation chemistry or electrochemical synthesis, can be used for various purposes. Moreover, various properties of PPy can be significantly improved by embedding, doping, or dedoping specific materials during or after its formation. PPy and its derivatives can be applied in enzyme-based biosensors, immunobiosensors, aptamer-based biosensors, MIP-based biosensors, and nanocatalytic biosensors and improve their sensing performance. Herein, it was shown that electrochemical-based PPy-based sensors, which have the inherent advantages of excellent sensitivity and selectivity, rapid detection, cost-effectiveness, and being capable of simultaneous detection of multiple CRC biomarkers, can be used to develop highly important detection strategies for CRC. However, the potential of such biosensors for simultaneous detection of multiple biomarkers needs further research and development. It is necessary to skillfully combine PPy with other nanomaterials to effectively improve the detection performance of biosensors. The stability of PPy during detection also needs to be further enhanced. In addition, more attention should be focused on improving the interfacial synergy between PPy and biorecognition elements, thereby improving the sensing performance of assays.

## Figures and Tables

**Figure 1 nanomaterials-13-00674-f001:**
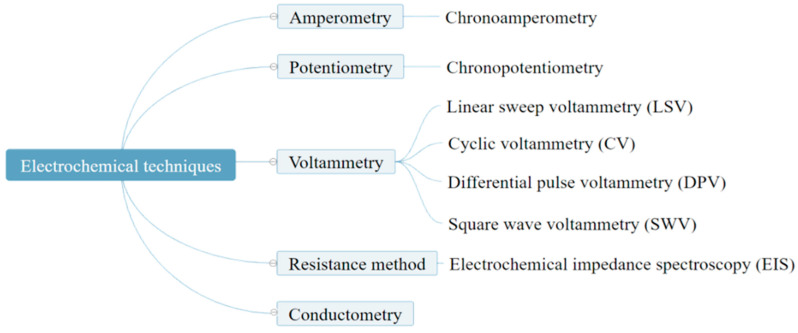
Electrochemical technologies commonly used in electrochemical detection.

**Figure 2 nanomaterials-13-00674-f002:**
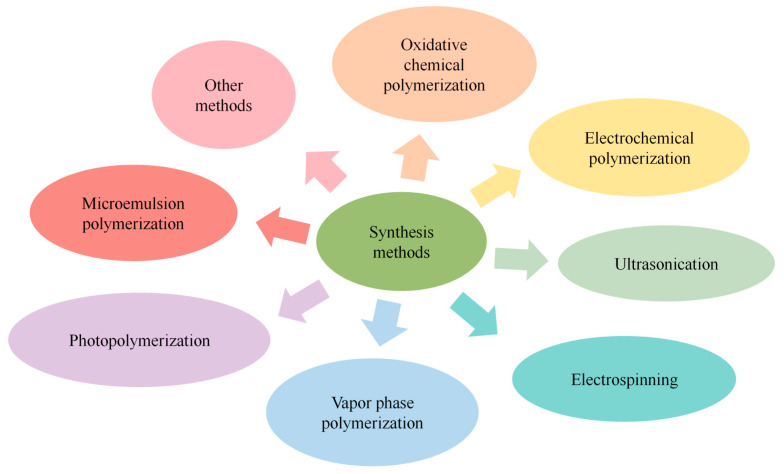
Synthesis methods for PPy.

**Figure 3 nanomaterials-13-00674-f003:**
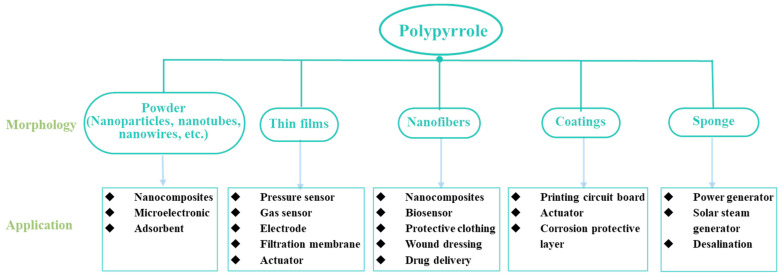
Synthesized PPy morphologies and their applications.

**Figure 4 nanomaterials-13-00674-f004:**
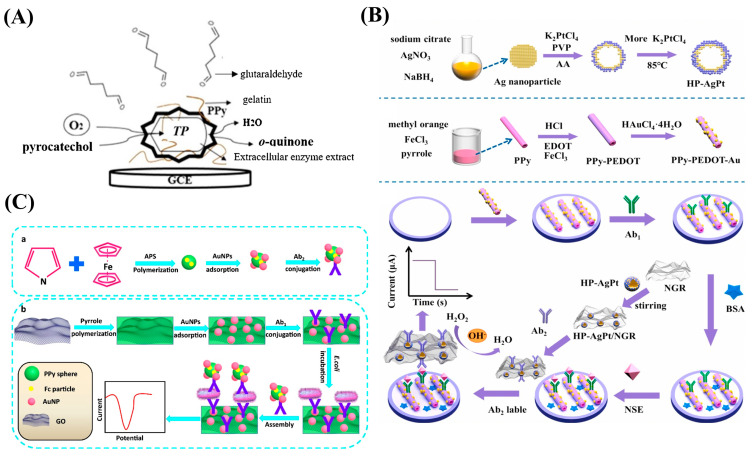
(**A**) Schematic diagram of PPy-based enzymatic sensor. Reproduced from [100] with permission from Elsevier. (**B**) Schematic diagram of the electrochemical platform for polypyrrole–poly(3,4-ethylenedioxythiophene)–gold (PPy-PEDOT-Au). Reproduced from [104] with permission from Elsevier. (**C**) Schematic diagram of the fabrication process for polypyrrole-reduced graphene oxide/gold nanoparticles (PPy-rGO/AuNPs) for use in biosensors. Reproduced from [105] with permission from Elsevier.

**Figure 5 nanomaterials-13-00674-f005:**
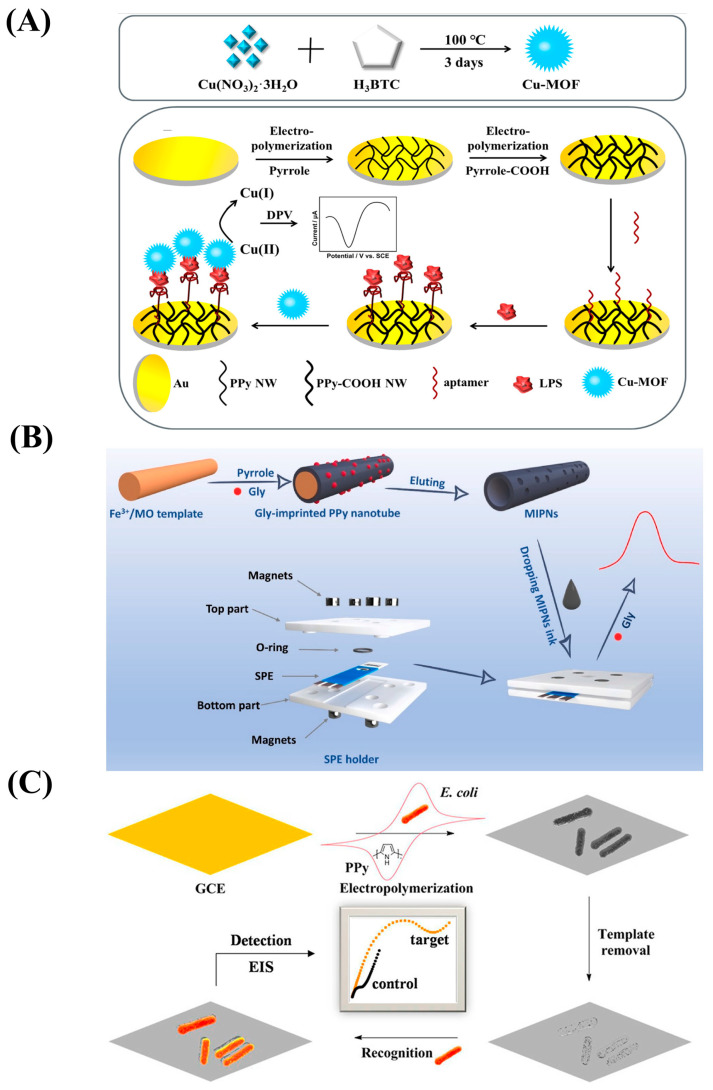
(**A**) Schematic diagram of the preparation procedure for the polypyrrole nanowire (PPyNW)-based aptasensor. Reproduced from [106] with permission from Elsevier. (**B**) Schematic diagram of the preparation of molecularly imprinted polypyrrole nanotubes (MIPNs) and the MIPN-based glyphosate platform. Reproduced from [108] with permission from Elsevier. (**C**) Schematic diagram of the fabrication process for the biosensor based on imprinted polypyrrole film from bacteria. Reproduced from [109] with permission from the American Chemical Society.

**Figure 6 nanomaterials-13-00674-f006:**
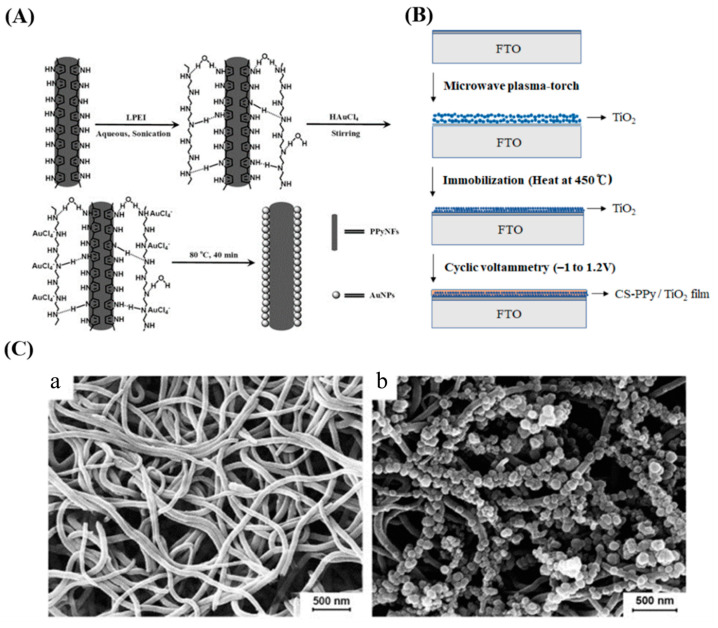
(**A**) Schematic diagram of the synthesis of polypyrrole nanofiber-supporting Au nanoparticles (Au/PPyNFs). Reproduced from [112] with permission from Elsevier. (**B**) Schematic diagram of the preparation of chitosan–polypyrrole/titanium oxide (CS-PPy/TiO_2_) nanocomposite films on a fluorine-doped tin oxide-coated glass slide (FTO). Reproduced from [113] with permission from MDPI. (**C**) Scanning electron microscope images of PPyNWs (**a**) and copper oxide (Cu_x_O) nanoparticle-modified PPyNWs (**b**). Reproduced from [114] with permission from Elsevier.

**Figure 7 nanomaterials-13-00674-f007:**
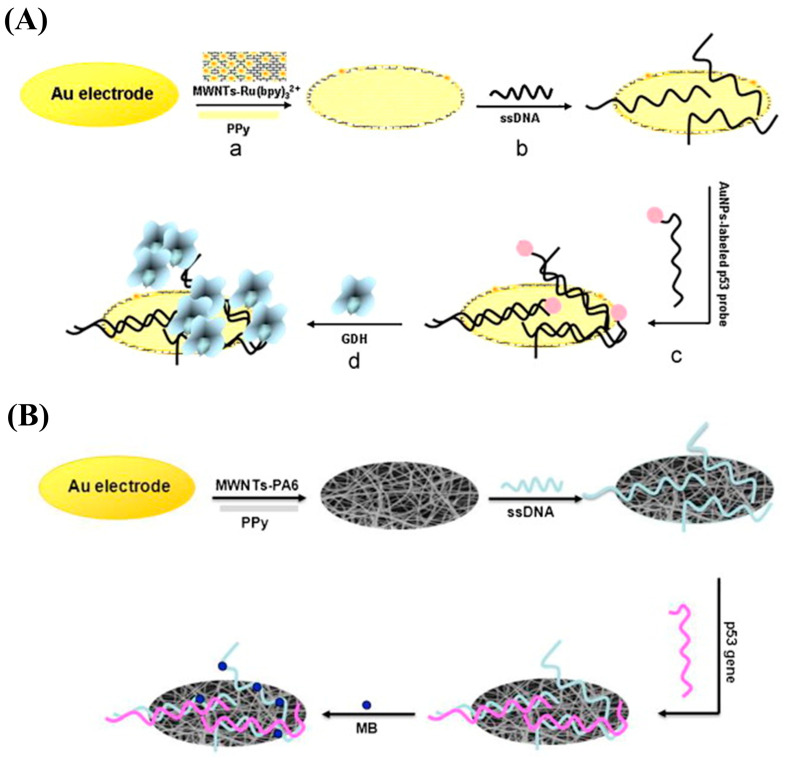
(**A**) Schematic illustration of the fabrication of a PPy-covered multiwalled carbon nanotube and ruthenium (II) tris-(bipyridine) (MWNT-Ru(bpy)_3_^2+^-PPy) biosensor for the wild-type p53 sequence (wtp53) assay based on electrochemical luminescence (ECL). Reproduced from [126] with permission from Elsevier. (**B**) Schematic illustration of the preparation of a multi-walled carbon nanotube–nylon 6–polypyrrole (MWNT-PA6-PPy) biosensor for wtp53 detection based on electrospinning technology. Reproduced from [127] with permission from Elsevier.

**Figure 8 nanomaterials-13-00674-f008:**
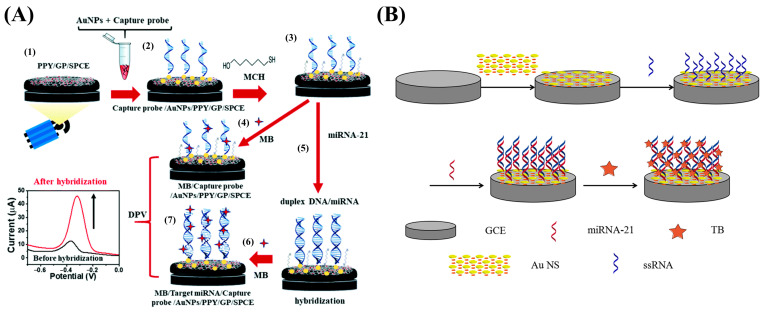
(**A**) Schematic representation of the strategy based on a gold nanoparticle/polypyrrole/graphene (AuNP/PPy/GP) nanocomposite. Reproduced from [139] with permission from the Royal Society of Chemistry. (**B**) Schematic representation of the approach using a PPy-AuNP superlattice (PPy-AuNS). Reproduced from [140] with permission from Elsevier.

**Figure 9 nanomaterials-13-00674-f009:**
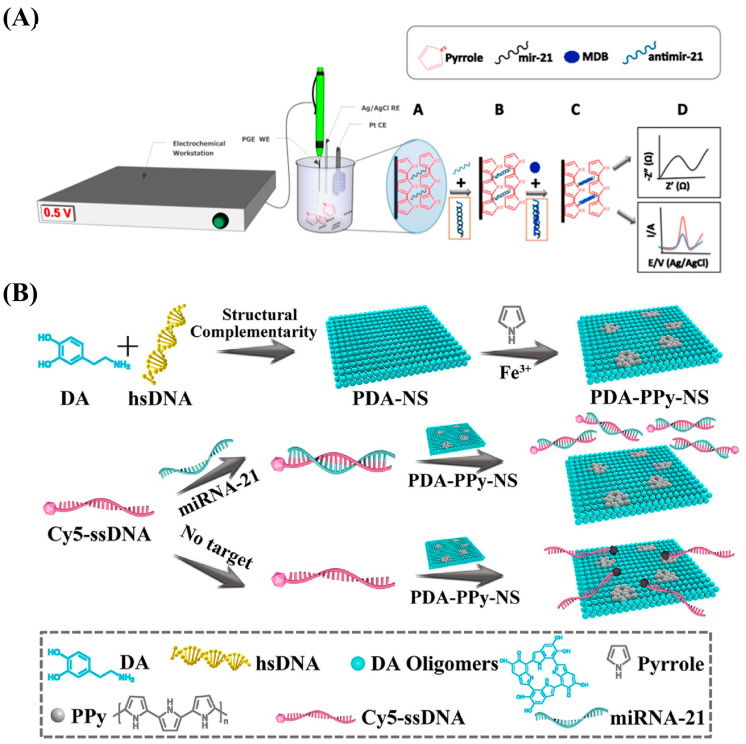
(**A**) Schematic diagram of the biosensor fabrication process utilizing PPy electrodeposition. Reproduced from [141] with permission from Elsevier. (**B**) Schematic diagram of the preparation procedure for the hybrid polydopamine/polypyrrole nanosheet (PDA-PPy-NS) biosensor for miRNA-21 determination. Reproduced from [83] with permission from the American Chemical Society.

**Figure 10 nanomaterials-13-00674-f010:**
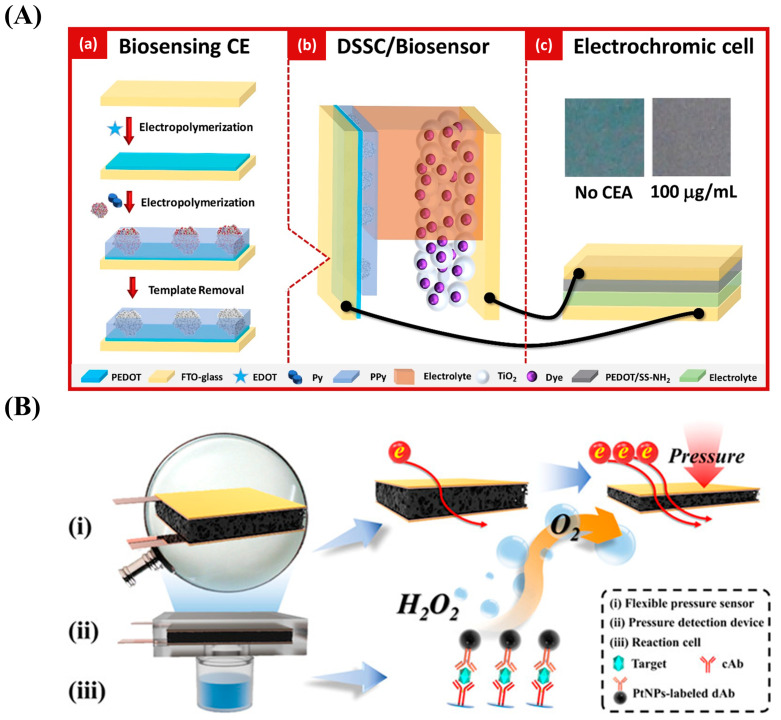
(**A**) Schematic diagram of the design of a self-powered and self-signaled biosensing platform based on biosensing, molecular imprinting technology, a dye-sensitized solar cell (DSSC), and electrochromic technology. Reproduced from [158] with permission from Elsevier. (**B**) Schematic diagram of a flexible pressure sensor based on the elastic three-dimensional (3D) structure of PPy foam. Reproduced from [143] with permission from the American Chemical Society.

**Figure 11 nanomaterials-13-00674-f011:**
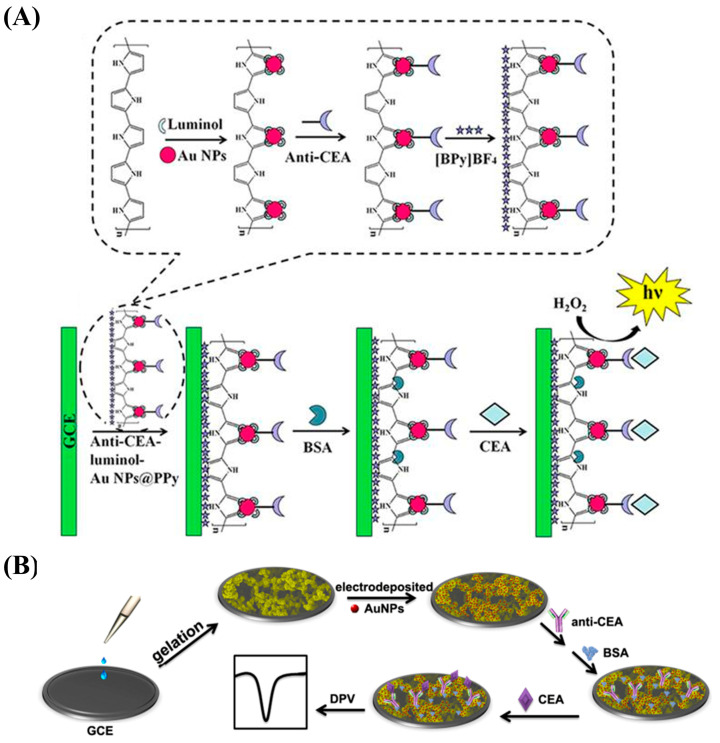
(**A**) Schematic diagram of the preparation of an anti-CEA–luminol–AuNP@PPy-based immunosensor for CEA detection. Reproduced from [144] with permission from Nature Publishing Group. (**B**) Schematic diagram of the electrochemical method for CEA determination based on 3D continuous conducting network nanocomposites composed of PPy hydrogel loaded with AuNPs. Reproduced from [147] with permission from Nature Publishing Group.

**Figure 12 nanomaterials-13-00674-f012:**
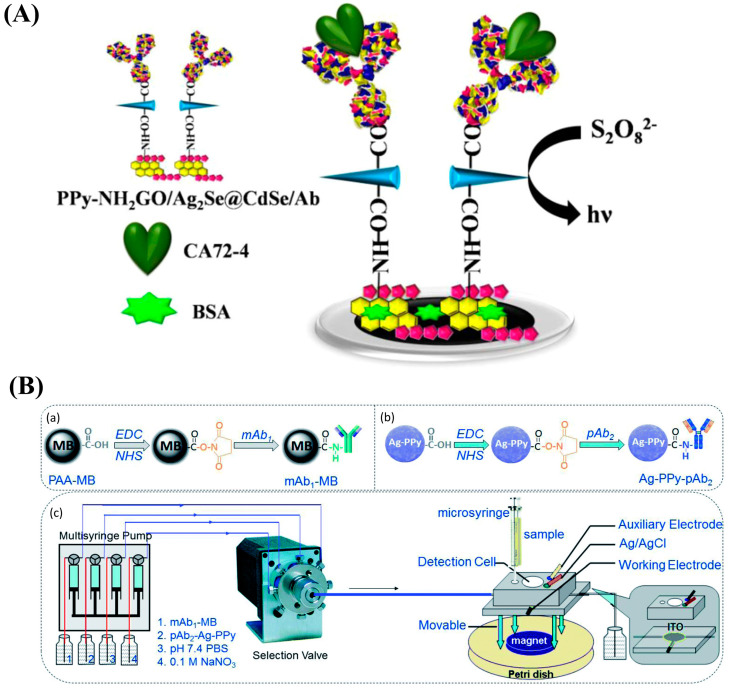
(**A**) Schematic diagram of the polypyrrole-intercalated aminated graphene/Ag_2_Se@CdSe (PPy-NH_2_GO/Ag_2_Se@CdSe)-based immunosensor for the CA72-4 assay. Reproduced from [148] with permission from the American Chemical Society. (**B**) Schematic diagram of the platform for CA125 measurement: (**a**) the preparation of the capture probe using mAb_1_-conjugated magnetic beads (mAb_1_-MB). (**b**) the preparation of the detection probe using pAb_2_-labeled Ag-PPy nanostructure (Ag-PPy- pAb_2_). (**c**) magneto-controlled microfluidic device with an electrochemical detection cell. Reproduced from [149] with permission from the Royal Society of Chemistry.

**Figure 13 nanomaterials-13-00674-f013:**
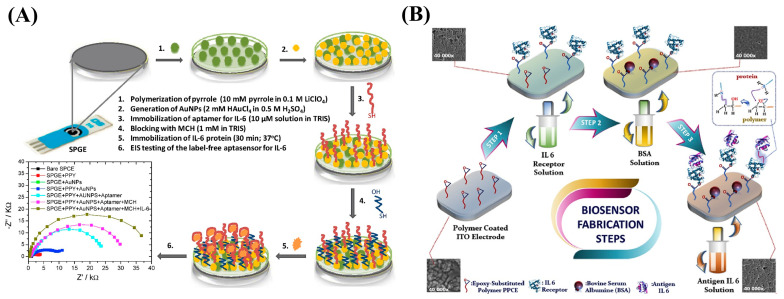
(**A**) Schematic diagram of the AuNP/PPyNP-based aptasensor for the IL-6 assay. Reproduced from [151] with permission from Elsevier. (**B**) Schematic diagram of the fabrication process for the PPCE-modified biosensor for IL-6 measurement. Reproduced from [152] with permission from Elsevier.

**Figure 14 nanomaterials-13-00674-f014:**
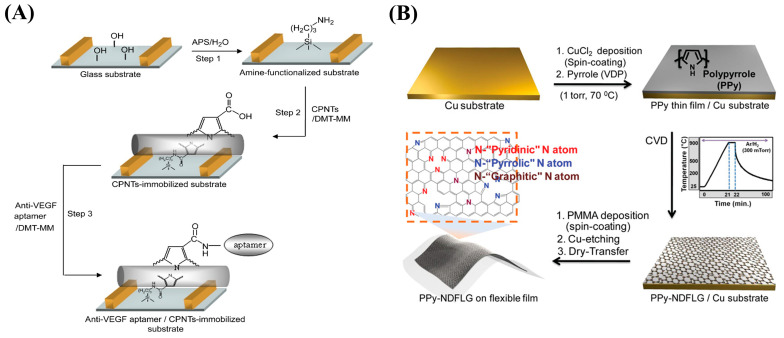
(**A**) Schematic diagram of the fabrication of a CPNT-modified aptasensor for the VEGF assay. Reproduced from [155] with permission from Elsevier. (**B**) Schematic diagram of the process of synthesizing flexible PPy-NDFLG. Reproduced from [156] with permission from the American Chemical Society.

**Table 1 nanomaterials-13-00674-t001:** Conductive polymers commonly used in electrochemical biosensors.

Conductive Polymers	Ref.
Polypyrrole (PPy)	[38,39,40,41]
Polyaniline (PANI)	[42,43,44]
Polythiophene (PTh)	[45,46,47]
Poly(3,4-ethylene dioxythiophene) (PEDOT)	[48,49,50]

**Table 2 nanomaterials-13-00674-t002:** Applications of PPy-based biosensors with different functions.

Type of Biosensor	Function of PPy	Ref.
Enzyme-based biosensors	♦ Entrap and immobilize the enzyme ♦ Reduce the oxidation potential of the substrate ♦ Improve the biocompatibility of the biosensor ♦ Promote electron transfer	[100,101,102]
Immunobiosensors	♦ Offer a proper environment for the immobilization of biomolecules ♦ Increase surface area ♦ Improve the biocompatibility and conductivity of the biosensor	[103,104,105]
Aptamer-based biosensors	♦ Immobilize aptamers ♦ Enlarge the specific surface area ♦ Enhance the biocompatibility and electroconductivity of the biosensor	[106,107]
MIP-based biosensors	♦ Manually create specific molecular recognition sites ♦ Enhance the specificity, biocompatibility, and electroconductivity of the biosensor	[108,109,110,111]
Nanocatalytic biosensors	♦ Serve as a suitable immobilization matrix and disperse metal nanoparticles effectively ♦ Improve the catalytic activity of enzyme mimics ♦ Provide good conductivity	[112,113,114]

**Table 3 nanomaterials-13-00674-t003:** PPy-based biosensors for the determination of CRC protein biomarkers.

Protein Biomarkers	Biosensor Components	Detection Method	Detection Range	LOD ^I^	Ref.
CEA ^II^	2-NS-PPy ^III^/PEE ^IV^-PPy/2-NS-PPy/AuNP/Apt/CEA	EIS	10^−1^–10^3^ ng/mL	0.033 ng/mL	[142]
	PPy foam/Cu/ITO ^V^/PET ^VI/^Kapton/PDMS ^VII^/cAb ^VIII^/CEA/PtNP-labeled dAb ^IX^	Resistance determination	0.2–60 ng/mL	0.13 ng/mL	[143]
	GCE ^X^/PPy@AuNP-luminol-anti-CEA/CEA	ECL	10^−5^–10 ng/mL	3 fg/mL	[144]
	ITO/PANI/PPy-Ag/Ab_1_/CEA/ZnO@AgNC ^XI^-Ab_2_	ECL	10^−3^–100 ng/mL	0.4 pg/mL	[145]
	AuNP/NH_2_-GS ^XII^/Ab_1_/CEA/Au@PdND ^XIII^/Fe^2+^-CS/PPy NT/Ab_2_	i-t/SWV	5 × 10^−5^–50 ng/mL	17 fg/mL	[146]
	GCE/PPy hydrogel/AuNP/anti-CEA/CEA	DPV	10^−6^–200 ng/mL	0.16 fg/mL	[147]
CA72-4 ^XIV^	GCE/PPy-NH_2_GO-Ag_2_Se@CdSe/Ab/CEA	ECL	10^−4^–20 U/mL	2.1 × 10^−5^ U/mL	[148]
CA125 ^XV^	ITO/MB-mAb_1_/CEA/PPy-Ag-pAb_2_	LSV	0.001–300 U/mL	7.6 mU/mL	[149]
	Au-SPE/MIPPy ^XVI^/CA125	SWV/SPR ^XVII^	0.01–500 U/mL	0.01 U/mL	[150]
IL-6 ^XVIII^	SPGE ^XIX^/PPy/AuPts/Apt ^XX^/IL-6	EIS	10^−6^–15 μg/mL	0.33 pg/mL	[151]
	ITO/PPCE ^XXI^/IL-6 receptor/IL-6	EIS/CV	0.02–16 pg/mL	6.0 fg/mL	[152]
	ITO/AB ^XXII^/EpxS-PPyr ^XXIII^/IL-6 receptor/IL-6	EIS/CV	0.01–50 pg/mL	3.2 fg/mL	[153]
	PEEK ^XXIV^/PETE ^XXV^/PPyNW/mAb/IL-6	EIS	1–50 pg/mL	0.36 pg/mL	[154]
VEGF ^XXVI^	Glass substrate/CPNT ^XXVII^/Apt/VEGF	FET ^XXVIII^	-	400 fM	[155]
	Flexible substrate/PPy-NDFLG ^XXIX^/Apt/VEGF	FET	-	100 fM	[156]

Notes: I. LOD: limitation of detection; II. CEA: carcinoembryonic antigen; III. 2-NS-PPy: PPy doped with 2-naphthalene sulfonate; IV. PEE: pentaerythritol ethoxylate; V. ITO: indium tin oxide; VI. PET: poly(ethylene terephthalate); VII. PDMS: poly(dimethylsiloxane); VIII. cAb: capture antibody; IX. dAb: detection antibody; X. GCE: glassy carbon electrode; XI. AgNC: silver nanocluster. XII. NH_2_-GS: amino-functionalized graphene sheet; XIII. PdND: palladium nanodendrite; XIV. CA72-4: carbohydrate antigen 72-4; XV. CA125: carbohydrate antigen 125; XVI. MIPPy: molecularly imprinted polypyrrole; XVII. SPR: surface plasmon resonance; XVIII. IL-6: interleukin-6; XIX. SPGE: screen-printed graphite electrodes; XX. Apt: aptasensor; XXI. PPCE: polypyrrole polymer-containing epoxy side group; XXII. AB: acetylene black; XXIII. EpxS-PPyr: epoxy-substituted poly(pyrrole) polymer; XXIV. PEEK: polyether ether ketone; XXV. PETE: poly(ethylene terephthalate); XXVI. VEGF: vascular endothelial growth factor; XXVII. CPNT: carboxylated polypyrrole nanotube; XXVIII. FET: field-effect transistor; XXIX. PPy-NDFLG: PPy-transformed N-doped few-layer graphene.

## Data Availability

Data available in a public (institutional, general or subject specific) repository that issues datasets with DOIs.

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
