# Peer review of "Application of Polypyrrole-Based Electrochemical Biosensor for the Early Diagnosis of Colorectal Cancer"

_nanomaterials, 2023, doi:10.3390/nano13040674_

Round 1

Reviewer 1 Report

The paper can be published after minor revision refleting comments inserted as yellow notes into attached pdf of submitted manuscript.

Author Response

Thank you very much for your kind comment. Your specific revision suggestions have been adopted including the abbreviations, inappropriate words and phrases in the text and quality of the figures, etc.

Reviewer 2 Report

Zhang et al. prepared a review article describing the application of polypyrrole-based electrochemical biosensors for the early diagnosis of colorectal cancer. Manuscript needs substantial improvement before consideration for publication. Any critical comments are missing. What readers/new researchers will grasp from this review is missing. My additional detailed comments are below.

1.  Figure 1, Commonly used electrochemical techniques? For PPy-based biosensors? Please revise the caption. Figure 2 is unnecessarily added.

2.      Table 1. It will be informative if Table 1, summarizing names of nanomaterials applied in biosensing, also includes references. Moreover, not to extend to other conducting polymers. The table should concentrate on polypyrrole and nanomaterials-based biosensors.

3.     Why ref. 34 is added? It described PANI application. The review title indicated that it would summarize papers describing polypyrrole-conducting polymer-based biosensors. Therefore, from my point of view, the author should concentrate on this topic, and any unnecessary references must be deleted.

4.      Figure 3 is taken from ref. 41 without mentioning copyright. Please check if you have any other figures/schemes added without taking proper copyright.

5.      Information added in Table 3 should be rechecked. Specially concentration ranges and LOD values. Once again, what are we summarizing/concluding from this Table? Nothing is discussed.

6.      Photopolymerization of pyrrole is not much studied. Author can emphasize such approaches which are not commonly used in synthesizing polypyrrole. Only listing papers can not be considered as a critical review. Moreover, if the author reports any uncommon approach, please cite the original article, not the review article describing it.

7.      Figure quality needs substantial improvement. Moreover, several figures are combined to make one figure. May be author can think of combining two such figures only, 4 figures in one Figure is too much. I could not see any labeling in figures.

8.      Figure 12 C? I cannot find an introduction/discussion of that figure.

9.   Opportunity and challenges. The title of the manuscript indicates that the review article will include a paper related to polypyrrole and enzymes. Therefore, selectivity in sensing should come from enzymes, not pyrrole.

10.   Conclusion is more suitable for Introduction. It needs improvement.

11.   I failed to see any takeaway massage from this review article. There is no critical discussion.

Author Response

  1. Why ref. 34 is added? It described PANI application. The review title indicated that it would summarize papers describing polypyrrole-conducting polymer-based biosensors. Therefore, from my point of view, the author should concentrate on this topic, and any unnecessary references must be deleted.

Answer: Thank you very much for your kind comment. Your recommendations are very pertinent and we have deleted the PANI and other unnecessary section to concentrate on the description of polypyrrole.

  1. Figure 3 is taken from ref. 41 without mentioning copyright. Please check if you have any other figures/schemes added without taking proper copyright.

Answer: Thank you very much for your kind comment. Figure 3 is our handmade figure with reference to ref. 41. We are very sorry for the confusion. We have cited this reference in the text instead of the caption part to make it clear to readers. What’s more, we make sure that we have taken proper copyright of other figures. 

  1. Information added in Table 3 should be rechecked. Specially concentration ranges and LOD values. Once again, what are we summarizing/concluding from this Table? Nothing is discussed.

Answer: Thank you very much for your kind comment. We have carefully rechecked the information added in Table 3 and the description of this table was added and marked in red color.

  1. Photopolymerization of pyrrole is not much studied. Author can emphasize such approaches which are not commonly used in synthesizing polypyrrole. Only listing papers can not be considered as a critical review. Moreover, if the author reports any uncommon approach, please cite the original article, not the review article describing it.

Answer: Thank you very much for your professional comment. Your suggestion is very much to the point. We have added the summary of photopolymerization of pyrrole including several other PPy polymerization methods in the revision version and marked in red color.

  1. Figure quality needs substantial improvement. Moreover, several figures are combined to make one figure. May be author can think of combining two such figures only, 4 figures in one Figure is too much. I could not see any labeling in figures.

Answer: Thank you very much for your suggestions. We have divided the figures, improved the figure quality and arranged the layout of pictures properly.

  1. Figure 12 C? I cannot find an introduction/discussion of that figure.

Answer: Thank you very much for your kind comment. We are very sorry for the mistake you point out. We discussed Figure 12C and Figure 12D in our previous edition, but the using "Figure 12A" and "Figure 12B" as a mistaken. It has been corrected in the revision version.

  1. Opportunity and challenges. The title of the manuscript indicates that the review article will include a paper related to polypyrrole and enzymes. Therefore, selectivity in sensing should come from enzymes, not pyrrole.

Answer: Thank you very much for your professional comment. We have discussed the application of PPy in biosensor for the increase of the stability, and the selectivity comes from enzyme. However, the discussion of interaction between polypyrrole and enzymes received much less attention, which we mentioned in the “Opportunity and challenges” part and marked in red color.

  1. Conclusion is more suitable for Introduction. It needs improvement.

Answer: Thank you very much for your kind comment. We have added more discussion of PPy-based biosensors to improve the conclusion.

  1. I failed to see any takeaway massage from this review article. There is no critical discussion.

Answer: Thank you very much for your suggestions. In the revision version, we have considered the whole article and added more critical evaluation, which is marked in red color.

Round 2

Reviewer 2 Report

Quality of manuscript is improved. However, still language need improvement.

Moreover, lies 173 and 180, need correction. PPy can not polymerize further because it is already polymerized.

Line 173, polypyrrole is already polymer, It should be "Py"

Line 180 PPy monomers...PPy is not monomer anymore, it is already polymerized

Author Response

Thank you very much for your kind comments and suggestions. Your suggestions are very much to the point, and we have adopted them and revised the manuscript. In addition, we further modify the language and grammar of the full text for the convenience of readers.